# LLaVA-Surg: Towards Multimodal Surgical Assistant via Structured Lecture Learning

## Abstract

Multimodal large language models (LLMs) have achieved notable success across various domains, while research in the medical field has largely focused on uni-modal images. Meanwhile, current general-domain multimodal models for videos still lack the capabilities to understand and engage in conversations about surgical videos. One major contributing factor is the absence of datasets in the surgical field. In this paper, we create a new dataset, Surg-QA, consisting of 102,000 surgical video-instruction pairs, the largest of its kind so far. To build such a dataset, we propose a novel two-stage question-answer generation pipeline with LLM to learn surgical knowledge in a structured manner from the publicly available surgical lecture videos. The pipeline breaks down the generation process into two stages to significantly reduce the task complexity, allowing us to use a more affordable, locally deployed open-source LLM than the premium paid LLM services. It also mitigates the risk of LLM hallucinations during question-answer generation, thereby enhancing the overall quality of the generated data. We further train LLaVA-Surg, a novel vision-language conversational assistant capable of answering open-ended questions about surgical videos, on this Surg-QA dataset, and conduct comprehensive evaluations on zero-shot surgical video question-answering tasks. We show that LLaVA-Surg significantly outperforms all previous general-domain models, demonstrating exceptional multimodal conversational skills in answering open-ended questions about surgical videos. We will release our code, model, and the instruction-tuning dataset.

## 1 Introduction

Surgery, as a discipline with rich multimodal information in the medical field, diverges significantly from general medical diagnoses that often depend on static imagery, such as magnetic resonance imaging and chest X-ray. The dynamic nature of surgical procedures with complex sequence of actions and multi-stage processes, cannot be fully captured or understood through a single image.

The medical field has recently witnessed the significant impact of the Large Language Model (LLM), especially in the arena of medical question answering. Domain-specific LLMs like LLaVA-Med (Li et al., 2023) and Med-PaLM (Singhal et al., 2022), fused with publicly accessible medical question-answer data such as PubMed (Zhang et al., 2023a), can assist with inquiries about a biomedical image and meet the safety-critical demands of the medical domain. Moreover, general purpose LLMs such as GPT (OpenAI, 2024), despite not being explicitly aligned to the medical field, have shown great potential and versatility when applied to some specific clinical knowledge areas. However, these models are still limited to processing single images, thus falling short of venturing into the surgical domain where the video modality plays a crucial role.

The availability of parallel video-text datasets has proven to be useful for pretraining generative model in a self-supervised manner, as demonstrated by conversational multimodal LLMs such as Video-ChatGPT (Maaz et al., 2023) and Video-LLaVA (Lin et al., 2023), and text-to-video generative models such as Sora (Brooks et al., 2024). However, obtaining surgical video-text pairs is more challenging than biomedical image-text pairs or general-domain video-text pairs due to the need of more expensive surgical expertise.

In this work, we introduce the **L**arge **L**anguage and **V**ision **A**ssistant for **Surg**ery (LLaVA-Surg), the first attempt at a surgical multimodal conversational assistant. LLaVA-Surg leverages an adapted LLM

that integrates the visual encoder of CLIP (Radford et al., 2021) with Llama (Touvron et al., 2023) as a language backbone, fine-tuned on generated instructional image-text pairs. Our approach further adapts the design for spatiotemporal video modeling and finetunes the model on video-instruction data to capture temporal dynamics and frame-to-frame consistency relationships available in video data.

A fundamental contribution of this work is the introduction of a novel two-stage question-answer generation pipeline. This pipeline extracts surgical knowledge from widely available surgical lecture videos, resulting in the creation of Surg-QA, a dataset comprising over 102K surgical video-instruction pairs. Each pair consists of a video and its corresponding instructional content in a question-answer format. This extensive and diverse dataset enables LLaVA-Surg's to understand surgical videos and engage in comprehensive conversations about surgical videos.

The major contributions of our paper are as follows:

1. *Surg-QA.* We introduce Surg-QA, to the best of our knowledge, the first large-scale surgical video instruction-tuning dataset, featuring over 102K surgical video question-answer pairs derived from more than 44K surgical video clips across 2,201 surgical procedures. We also introduce the novel two-step question-answer generation pipeline behind Surg-QA. This pipeline effectively mitigates the issue of LLM hallucination, providing a cost-effective solution for large-scale question-answer generation.

2. *LLaVA-Surg.* We present LLaVA-Surg, to the best of our knowledge, the first video conversation model capable of expert-level understanding of surgical videos and answering open-ended questions about surgical videos. LLaVA-Surg is trained in under 6 hours using eight A100 GPUs, by fine-tuning a general-domain vision-language model on Surg-QA. Comprehensive evaluations show that LLaVA-Surg excels in zero-shot surgical video question-answering tasks, outperforming previous models and demonstrating strong multimodal conversational skills.

3. *Open-source.* We will publicly release the surgical video instruction-tuning dataset, model, and code for data generation and training to advance research in the surgical domain.

## 2 RELATED WORK

**Surgical Video Question Answering (Surgical VQA)** models can answer questions based on surgical videos and offer assistance to practicing surgeons and surgical trainees. Early surgical VQA methods were largely discriminative (Twinanda et al., 2016; Czempiel et al., 2020; Yengera et al., 2018), treating the task as a classification problem where answers were chosen from a predefined set. They excelled in identifying surgical steps, instruments, and organs, but were limited to closed-set predictions and struggled with open-ended questions and answers. Recent developments have shifted towards generative methods (Seenivasan et al., 2022; Bai et al., 2023; Seenivasan et al., 2023) that produce free-form text sequences but are limited to single-turn conversations, preventing them from engaging in a dialogue or answering follow-up questions. Unlike these models, our LLaVA-Surg model can engage in meaningful multi-turn dialogues, answering surgical questions and providing comprehensive surgical knowledge for an interactive learning experience.

**Multimodal LLM for Biomedical Image Conversations** represents a significant advancement in the field of medical artificial intelligence. These models combine text and image understanding to enable more nuanced and contextually aware interactions between clinicians and AI systems. For instance, the LLaVA-Med model demonstrates the potential of multimodal LLMs to interpret and generate detailed medical image descriptions, thereby aiding both diagnostics and patient communication (Li et al., 2023). The application of such models extends to various tasks including VQA, where they provide accurate and relevant answers based on medical images and related queries (Zhang et al., 2023b; Pal et al., 2023). This multimodal approach also enhances the ability to perform complex reasoning and decision-making processes, which are critical in clinical settings (Liu et al., 2024a). Collectively, these developments underscore the transformative potential of multimodal LLMs in enhancing biomedical image conversations and ultimately improving patient care outcomes (He et al., 2020; Lau et al., 2018).

**Multimodal LLM for Video Conversations** has demonstrated great potential by integrating general-domain text, images, and video data. Early works like FrozenBiLM (Yang et al., 2022) demonstrates

the promise of aligning vision and language models for multimodal understanding. Recent advancements like Video-LLaVA (Lin et al., 2023), Video-ChatGPT (Maaz et al., 2023), and ChatUniVi (Jin et al., 2024) illustrate practical applications in video contexts, delivering real-time, contextually aware responses that improve user interactions. Specifically, Video-LLaVA integrates visual and language data using the Language-Bind framework, enhancing video understanding and generating coherent, contextually relevant responses. Video-ChatGPT excels in handling complex video data, providing detailed analysis and responses. ChatUniVi pushes the boundaries further by integrating unified video and language processing capabilities, facilitating more natural and interactive video conversations. But their applicability to domain-specific videos like surgery videos have not yet been proven.

## 3 SURGICAL VIDEO INSTRUCTION-TUNING DATA GENERATION

There is a significant deficiency in specialized datasets for training multimodal LLM as a conversational assistant in the surgical domain. As illustrated in Figure 1, information in the surgical domain can be categorized into four distinct levels: (1) basic identification of surgical objects such as organs and instruments, (2) recognition of discrete surgical actions, (3) higher-order reasoning of surgical actions, and (4) expert level deduction and planning.

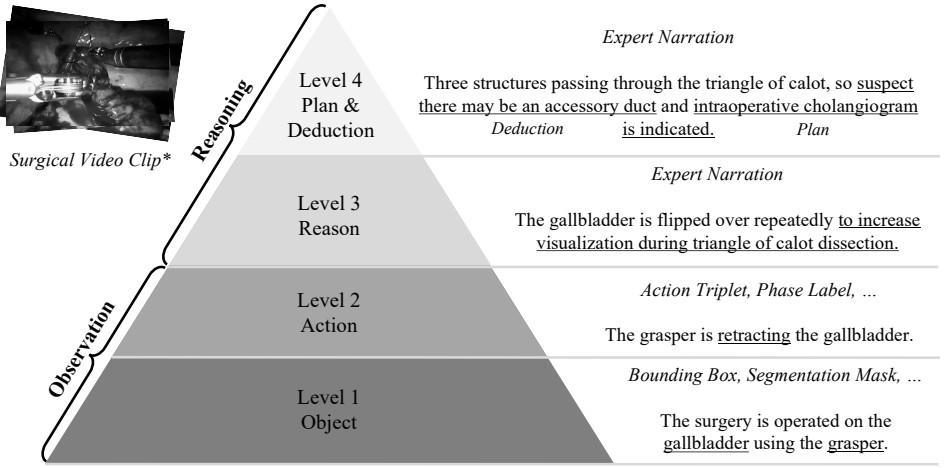

* Video frames are colorful in our dataset. We convert them into grayscale here for the sake of readability.

Figure 1: Surgical Knowledge Pyramid. Surgical video interpretation can be categorized into four levels. The first two levels represent the observation capabilities, which can be captured by traditional computer vision tasks such as object detection, segmentation, and labeling. But this only conveys a superficial level of understanding. The next two levels represent the reasoning capabilities. Interpretation at the reasoning levels provides the rationale behind the observations, further offering deductions and plannings, conveying deep, surgical expert-level understanding.

However, existing datasets (Bai et al., 2023; Yuan et al., 2024) lack level 3 and 4 information. To address this, we create *Surg-QA*, the first surgical instruction-tuning dataset that contains all four levels of information. The proposed dataset consists of 100K video-text pairs from structured learning of surgical lecture videos and 2K pairs focusing on the surgical visual concept alignment.

**Surgical Video Instruction-Tuning Data.** For a surgical video $\mathbf{X}_v$ and its transcript $\mathbf{X}_t$, we prompt Llama-3-70B (AI, 2024) through a two-step approach to create a set of questions $\mathbf{X}_q$ that can be answered only when the video is provided, aiming to guide the assistant in describing the video content. A single-round instruction-tuning example can thereby represented by:

$$\texttt{User} : \mathbf{X}_q \, \mathbf{X}_v \texttt{<STOP>} \backslash \text{n} \; \texttt{Assistant} : \mathbf{X}_a \texttt{<STOP>} \backslash \text{n} \tag{1}$$

**Structured Surgical Video Learning.** We propose a two-step *extraction-generation* approach utilizing the Llama-3-70B model for processing surgical video lectures, as illustrated in Figure 2. Specifically, given a surgical lecture video $\mathbf{X}_v$ with voiceover, we begin by applying WhisperX (Bain et al., 2023) to transcribe the spoken content of surgical lecture videos into text. Following this,

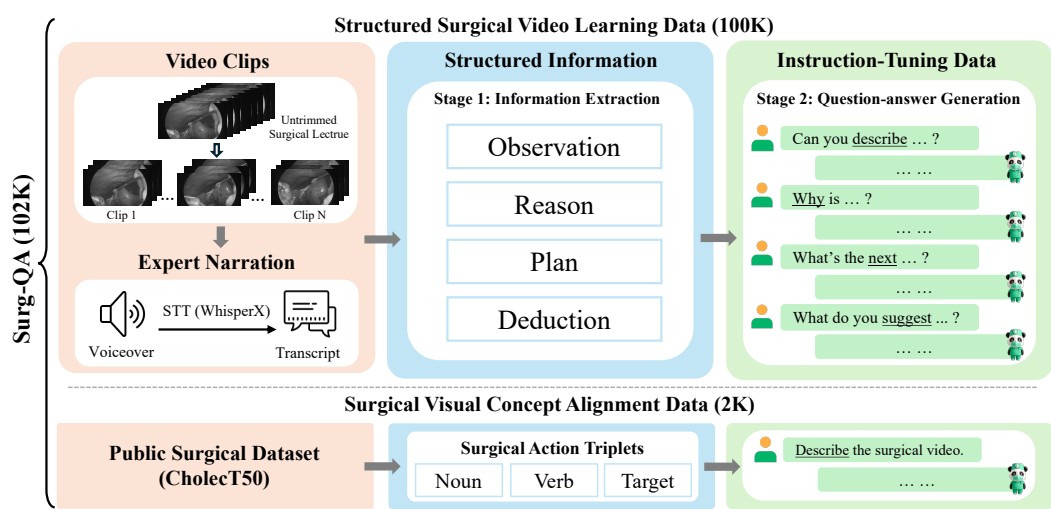

Figure 2: Instruction-Tuning Data Generation Pipeline. Top: Structured surgical video learning begins with untrimmed lecture videos divided into clips. Expert narrations (transcripts) from the lectures are converted to text using WhisperX Bain et al. (2023). We then prompt Llama-3-70B to extract the structured information from the transcripts. Finally, the extracted information is provided to Llama-3-70B to generate the instruction-tuning data. Bottom: Surgical visual concept alignment data are concise descriptions of surgical videos, generated based on surgical action triplets.

unlike previous work (Gilardi et al., 2023; Liu et al., 2024b; Li et al., 2023) that directly prompt LLM to generate multi-round questions and answers based on the text information, we first prompt LLM to extract the key information from the transcripts in a structured manner, focusing on four main components: the observation $\mathbf{I}_o$ and the corresponding reason $\mathbf{I}_r$, plan $\mathbf{I}_p$ and deduction $\mathbf{I}_d$ as shown in Figure 1. This structured representation of videos ensures high-quality data by extracting only surgery-related information, thus mitigating noise from non-surgical clips or non-informative conversations. Additionally, it reduces the risk of LLM hallucination (Huang et al., 2023; Li et al., 2023) by restricting the model to information extraction only. We also manually curate few-shot examples to teach how to extract high-quality information based on the transcript. See Appendix A.2 for the prompt and few-shot examples.

Once the information has been extracted, we can create the instruction-tuning data as multi-turn conversations by prompting LLM to generate different types of question-answering pairs in a controllable way. For example, by concatenating all the observations $(\mathbf{I}_o^1, \mathbf{I}_o^2, \ldots, \mathbf{I}_o^T)$ where $T$ is the total observations of $\mathbf{X}_v$, we prompt LLM to generate the first question-answer pair $[\mathbf{X}_q^1, \mathbf{X}_a^1]$ that focus on the visual content of the surgical lecture. Next, for each of the $[\mathbf{I}_o, \mathbf{I}_r]$, $[\mathbf{I}_o, \mathbf{I}_p]$ and $[\mathbf{I}_o, \mathbf{I}_d]$ combinations, we prompt LLM to generate the surgical reasoning question-answering pairs $(\mathbf{X}_q^2, \mathbf{X}_a^2, \ldots, \mathbf{X}_q^N, \mathbf{X}_a^N)$ where $N$ is the total number of question-answer pairs. By stacking the question-answer pairs, we can create a multi-turn conversation, where the instruction $\mathbf{X}_q^t$ at the $t$-th turn is defined as:

$$\mathbf{X}_{\mathbf{q}}^t = \begin{cases} [\mathbf{X}_q^1, \mathbf{X}_v] \text{ or } [\mathbf{X}_v, \mathbf{X}_q^1], & t = 1 \\ \mathbf{X}_q^t, & t > 1 \end{cases} \quad (2)$$

We can then construct the multi-turn multimodal instruction-tuning data:

$$\texttt{User}: \mathbf{X}_{\mathbf{q}}^1\, \mathbf{X}_v \texttt{<STOP>} \backslash \texttt{n Assistant}: \mathbf{X}_{\mathbf{a}}^1 \texttt{<STOP>} \backslash \texttt{n}$$
$$\texttt{User}: \mathbf{X}_{\mathbf{q}}^2\, \texttt{<STOP>} \backslash \texttt{n Assistant}: \mathbf{X}_{\mathbf{a}}^2 \texttt{<STOP>} \backslash \texttt{n} \ldots \ldots \quad (3)$$

An example of instruction-tuning data is shown in Figure 3. In comparison, we provide the pairs generated with the same information using the previous end-to-end approach (Li et al., 2023; Liu et al., 2024b), the previous approach generated an incorrect pair due to the hallucination. The prompt for structured information extraction is provided in Appendix A.2.

We collected 2,151 surgical lecture videos from WebSurg[1] (WebSurg, 2024). As shown in Figure 4c, these videos cover upper and lower gastrointestinal, hepatobiliary, urologic, gynecologic, general

---

[1] https://www.websurg.com

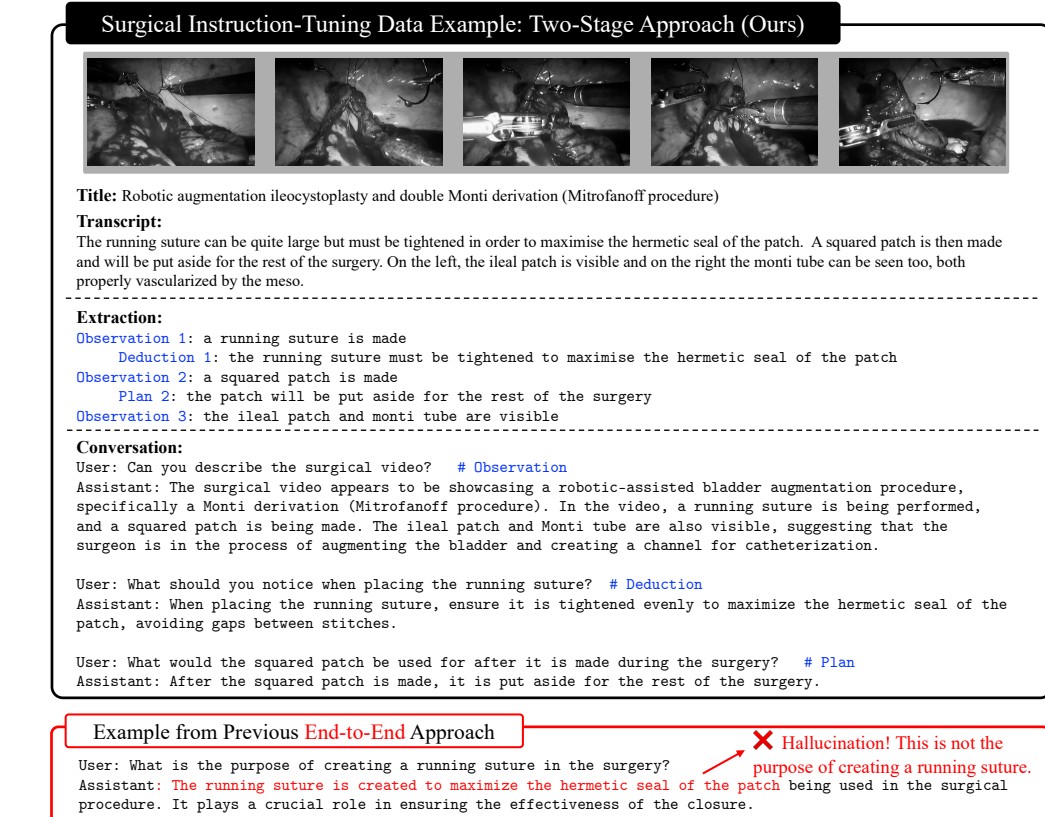

Figure 3: Comparison of instruction-tuning data generated by our two-stage approach (top) and the previous end-to-end approach (bottom). Both approaches were given the same video title and transcript. Our approach accurately extracted information from the transcript, generating correct question-answer pairs. In contrast, the conventional end-to-end approach produced incorrect question-answer pairs due to hallucination.

hernia, pediatric, endocrine, solid organ, and thoracic surgeries. We divided them into 42K short clips (15-30 seconds). Our automated pipeline generated 100K video-text pairs. We provided detailed statistics of Surg-QA in Figure 4.

**Surgical Visual Concept Alignment.** We create the surgical visual concept alignment data based on the public surgical dataset CholecT50, which aids the model in recognizing fundamental surgical visual concepts such as instruments, organs, and actions. CholecT50 includes 50 endoscopic videos, each frame annotated with action triplets: $[\texttt{instrument}, \texttt{verb}, \texttt{target}]$ that denote the tool, action, and the object or site of the action, respectively. We first divide the videos into 30-60-second clips. To generate a concise description for each video clip, we begin by merging consecutive frames with the same annotations while preserving the chronological order. Once this sequence of merged annotations is obtained, we use the sequence to prompt a Llama-3-70B to generate a description of the clip. In total, we sampled 2,200 video-text pairs to create the instruction-tuning dataset as outlined in Equation 1.

**Comparisons.** We compare Surg-QA with both existing general-domain VQA datasets and surgical-domain VQA datasets as shown in Tables 1 and 2. First, regarding whether Surg-QA is sufficient to train a multimodal LLM: Table 1 demonstrates that Surg-QA is substantial in size, with 44K videos and 102K QA pairs, making it comparable to general-domain VQA datasets. Second, Surg-QA surpasses traditional surgical-domain VQA datasets. As shown in Table 2, Surg-QA includes more surgical procedures, and a wider range of surgical types (Figure 4c), and provides video-wise

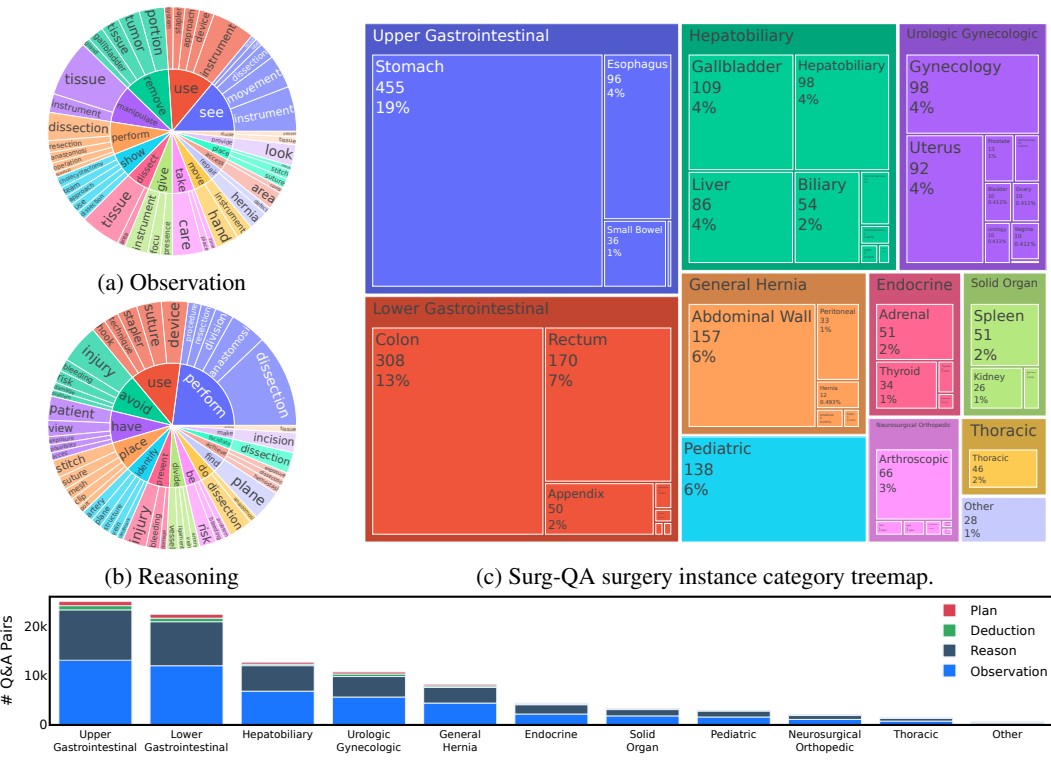

(a) Observation

(b) Reasoning

(c) Surg-QA surgery instance category treemap.

(d) Frequencies of QA pairs by categories.

Figure 4: The data statistics of surgical multimodal instruction-tuning data: (a,b) The root verb-noun pairs provide an overview of our dataset of instructions and responses. In the plot, the inner circle represents the root verb of the response, and the outer circle represents the direct nouns. (c) The distribution of videos of different types. (d) The distribution of video and QA pairs on 11 categories.

Table 1: Comparison with existing general-domain VQA datasets.

| General VQA Datasets | Q&A pairs generation | # Video clips | # Q&A pairs | Avg. length |
|---|---|---|---|---|
| MSVD-QA Xu et al. (2017) | Automatic | 2K | 51K | 10s |
| ActivityNet-QA Yu et al. (2019) | Human | 6K | 60K | 180s |
| MovieQA Tapaswi et al. (2016) | Human | 7K | 7K | 200s |
| MSRVTT-QA Xu et al. (2017) | Automatic | 10K | 244K | 15s |
| VideoInstruct-100K Maaz et al. (2023) | Human&Automatic | – | 100K | - |
| Surg-QA (Ours) | Automatic | 44K | 102K | 20s |

Table 2: Comparison with existing surgical-domain VQA datasets.

| Surgical VQA Dataset | # Surgical procedures | Total length (Hour) | Video-wise Q&A | Knowledge Observation | Reasoning |
|---|---|---|---|---|---|
| EndoVis-18-VQA Seenivasan et al. (2022) | 14 | – | ✗ | ✓ | ✗ |
| Cholec80-VQA Seenivasan et al. (2022) | 80 | 24 | ✗ | ✓ | ✗ |
| SSG-VQA Yuan et al. (2024) | 40 | 28 | ✗ | ✓ | ✗ |
| Surg-QA (Ours) | 2201 | 233 | ✓ | ✓ | ✓ |

question-answer pairs rather than frame-wise annotations. It also integrates both observational and reasoning-based knowledge, offering a comprehensive understanding of surgical procedures.

## 4 SURGICAL VISUAL INSTRUCTION TUNING

**Architecture.** LLaVA-Surg is a large vision-language model that aims to generate meaningful conversation about surgical videos. It employs the architecture of Video-ChatGPT (Maaz et al., 2023), a general-domain multimodal conversation model. Given a video, the model first samples $N$ frames

uniformly, and calculate the frame-level features $h \in \mathbb{R}^{N \times h \times w \times D}$ for each of the frames using CLIP ViT-L/14 (Radford et al., 2021), where $D$ is the hidden dimension of CLIP features and $h, w$ are the video height and width respectively. The features $h$ are fused through a temporal-fusion operation, where the temporal features $t \in \mathbb{R}^{N \times D}$ are derived through an average-pooling operation along the temporal dimension, and spatial features $s \in \mathbb{R}^{(h \times w) \times D}$ are derived using the same average-pooling operation but along the spatial dimensions. By concatenating $t$ and $s$, we derived the video-level features $f \in \mathbb{R}^{(N+h \times w) \times D}$, then feed it through a linear projection layer that connects $f$ to the language model.

**End-to-End Instruction-Tuning.** To balance the knowledge from levels 1 to 4, we combine the structured surgical video learning data and concept alignment data as discussed in Section 3, this results in 38K training video clips with 90K question-answer pairs. These pairs are converted to instruction-following data as described in Equation 3, the data includes instructions that simply present the task of describing the video, and tasks that answer various reasoning tasks. To train the model to follow various instructions and complete tasks in a conversational manner, we finetune LLaVA-Surg as a chatbot on the conversational data. During our training, we keep the weights of the CLIP visual encoder only and finetune the rest of the parameters.

## 5 EXPERIMENTS

We conduct experiments to study two key components: the performance of LLaVA-Surg and the quality of the produced multimodal surgical instruction-tuning data. Our experiments focus on two evaluation settings: (1) How does LLaVA-Surg perform in surgical video question-answering, and how does it compare to existing methods in the surgical domain? (2) How does the GPT evaluation framework compare to the clinician evaluation?

### 5.1 IMPLEMENTATION DETAILS

**Data.** We collected 2,054 surgical procedures from WebSurg using the keyword "intervention" and an additional 97 procedures with the keyword "gallbladder" for future evaluation purposes, totaling 2,151 procedures. These were randomly divided into a training set of 1,935 procedures and a test set of 216 procedures. In our instruction-tuning data generation pipeline, we use the 'large-v2' version of WhisperX (Bain et al., 2023) to transcribe the surgical lectures. We use Llama-3-70B-Instruct (AI, 2024) for information extraction and data generation as mentioned in Section 3. We use 'gpt-3.5-turbo-0125' to perform the following quantitative evaluation.

**Training.** We use LLaVA-Med as our pre-trained language backbone and finetune the model on 90K surgical video instruction following data. We use CLIP ViT-L/14 as the image encoder and use LLaVA-Med's language backbone as the initial weight of LLaVA-Surg. We update the linear layer projecting the video features to the LLM's input space and the language backbone, while the CLIP encoder is kept frozen. We finetune the model for 5 epochs using a learning rate of 2e-5 and an overall batch size of 128. The training of our 7B model took around 6 hours on 8 A100 40GB GPUs. For the rest of the hyperparameters, we follow the settings in (Maaz et al., 2023).

### 5.2 QUANTITATIVE EVALUATION

Table 3: Comparison of Zero-shot Surgical Question-Answering on Surg-QA.

| Model | Score (0-5) | Accuracy@all | Accuracy@1 |
|---|---|---|---|
| LLaVA-Med | 1.30 | 0.123 | 0.211 |
| Video-LLaVA | 1.32 | 0.129 | 0.224 |
| Video-ChatGPT | 1.04 | 0.098 | 0.172 |
| LLaVA-Surg (Ours) | **2.45** | **0.308** | **0.545** |

**Question-Answer Evaluation.** We conducted a comprehensive quantitative evaluation on the test split of Surg-QA consisting of 4359 open-ended surgical video question-answer pairs. Following recent works (Lin et al., 2023; Maaz et al., 2023; Li et al., 2023) that use GPT to evaluate open-ended

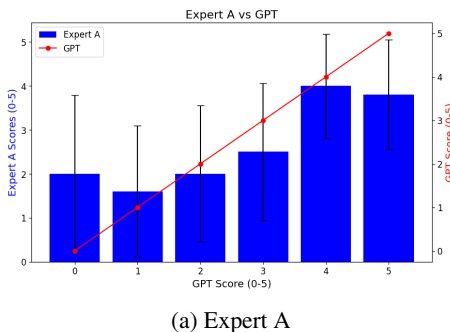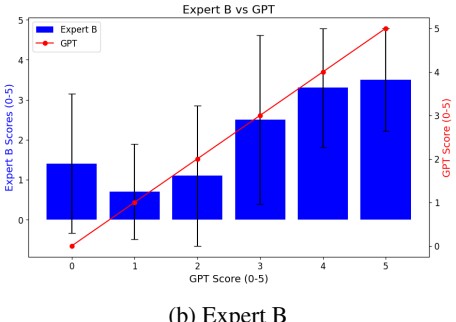

(a) Expert A             (b) Expert B

Figure 5: Clinician Evaluations vs GPT Evaluation. We conducted clinician evaluation experiments with two experts, A (a) and B (b), to assess LLaVA-Surg's responses to 60 surgical videos. The results from both experts demonstrate that the evaluations provided by GPT are comparable to those conducted by clinicians, affirming the reliability of GPT's assessment in this context.

questions, our evaluations employ GPT-3.5-Turbo for evaluation to assess the model's capabilities of answering surgical video questions. This evaluation process measures the accuracy of the model's generated predictions and assigns a relative score on a scale from 0 to 5. We provide the prompt used for evaluation in Appendix A.2.

In our evaluation process, GPT-3.5-Turbo was utilized to score the model's outputs by comparing them with the ground truth from the dataset. Each output was rated on a scale from 0 to 5 based on how accurately it reflected the observations. This approach enables us to directly determine the accuracy of the model's predictions. To achieve this, we provided GPT with the extracted observations as mentioned in Section 3, allowing it to evaluate the correctness of the observations included in the answers. Additionally, GPT-3.5-Turbo offered detailed comments highlighting the matches and discrepancies for further reference. Our results are presented in Table 3, where we provide the GPT evaluation scores. Additionally, we calculated the accuracy when at least one observation is matched (accuracy@1) and the overall accuracy for all observations in the test set (accuracy@all).

To benchmark LLaVA-Surg, we compared its performance with other significant models such as Video-LLaVA and Video-ChatGPT. Despite the solid foundation established by these models, LLaVA-Surg outperformed them in the surgical domain, achieving state-of-the-art (SOTA) performance. We also compare with LLaVA-Med which is an MLLM in the biomedical image domain that supports only unimodal images, we feed the first frame of the video clip into the model, and the results demonstrate the importance of video modality to the surgical domain. These results indicate LLaVA-Surg's ability to understand the surgical video content and generate accurate, contextually rich answers to questions.

**Clinician Evaluation.** To validate whether the GPT evaluation framework can benchmark the model's true performance, two clinicians are asked to evaluate a subset of the experimental results. The surgeon assigned a score from 0 to 5 to LLaVA-Surg's response based solely on his understanding of the surgical videos. We also provided him with the prompt used for GPT evaluation as a reference. Specifically, we randomly sampled a portion of data for each score from the GPT evaluation results, resulting in a total of 60 video-text pairs, then these samples were compared with the average scores given by the clinicians for each score. We report the comparison results between clinician evaluation and GPT evaluation in Figure 5. The figure reveals a strong correlation between the scores assigned by the clinician and GPT. This similarity in the scoring patterns indicates that the GPT evaluation framework effectively aligns with clinician judgment and offers a more scalable solution for evaluation. The full human evaluation results with comments are provided in the supplementary materials.

## 5.3 QUALITATIVE EVALUATION

We performed an extensive evaluation of our model on various open-ended surgical video question-answering tasks. Table 4 illustrates an example involving a gastric bypass procedure. LLaVA-Surg accurately identifies the procedure as a gastric bypass, noting the use of a suture and the

| Surgical Video Question-answer Example | |
| --- | --- |

Video Source (583s to 599s): https://websurg.com/en/doi/vd01en2844/

| User | Can you describe the surgical video? |
| --- | --- |
| LLaVA-Surg | The surgical video appears to be showcasing a laparoscopic Roux-en-Y gastric bypass procedure. In the video, we see the surgeon performing a suture, which suggests that the procedure is in its later stages, possibly during the closure of the jejunostomy or gastrojejunostomy. The camera provides a clear view of the surgical site, allowing us to observe the intricate movements of the instruments and the surgeon's hands as they work to complete the bypass. |
| User | How is the petersen space closed in this procedure? |
| LLaVA-Surg | The petersen space is closed using a non-absorbable monofilament suture. |
| User | Can you describe the surgical video? |
| LLaVA-Med | The image is a surgical video showing the exterior view of a grossly macroscopically normal appendix. This means that the appendix appears normal and unobstructed, and it is visible to the naked eye. The video provides a visual representation of the appendix's appearance during the surgical procedure. |
| User | How is the petersen space closed in this procedure? |
| LLaVA-Med | In this procedure, the Petersen space is closed using an end-to-end stapler. This is a common technique used to close the space after the removal of an appendix or other tissue. |
| User | Can you describe the surgical video? |
| Human Expert | This appears to be a bypass procedure for weight loss. The small bowel and mesentery are visible, along with a laparoscopic needle driver, needle and monofilament suture. The surgeon is closing Peterson's space which was created after dissection and creation of the smallbowel anastmosis, with a running suture in order to prevent the complication of internal hernia. |
| User | How is the petersen space closed in this procedure? |
| Human Expert | Peterson's space is closed with running monofilament suture in this clip. |
| User | Can you describe the surgical video? |
| Llama-3-70B (Language only) | This surgical video appears to be showcasing a complex weight loss surgery, specifically a gastric bypass procedure. In the video, we see the surgeon carefully closing a specific area, likely to prevent internal herniation, and using a purse string technique to secure the tissue. |
| User | How is the petersen space closed in this procedure? |
| Llama-3-70B | A purse string is performed to correctly close this space using non-absorbable suture material. |

Table 4: Example comparison of surgical video question-answering. We provided the ground truth answers generated by the language-only Llama-3-70B for reference. The answers are based solely on extracted information and the video title. It is considered the model's performance upper bound.

closing operation. It correctly answers the subsequent question regarding using a non-absorbable monofilament suture to close the Petersen space. However, LLaVA-Med fails to correctly describe the video, nor answer the following question. We provide more examples in Appendix B.

# 6 CONCLUSION

In this paper, we introduced Surg-QA, a surgical video instruction-tuning dataset of 102K video-text pairs. Surg-QA is generated primarily through a cost-efficient, two-stage question-answer generation pipeline, which effectively reduces hallucinations during question-answer generation by LLM. We then trained LLaVA-Surg, a multimodal LLM in the surgical video domain, on Surg-QA. LLaVA-Surg shows great potential in understanding surgical videos and engaging in surgical video conversations, outperforming previous multimodal LLMs in our comprehensive evaluation. While LLaVA-Surg performs competitively compared to existing methods in the surgical video domain, we note that LLaVA-Surg is limited by hallucinations. Future work is directed toward engaging experts to review the generated samples in Surg-QA to improve the accuracy and reliability of LLaVA-Surg.

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

## A DATA

### A.1 SURG-QA

We open-source the surgical instruction-tuning dataset Surg-QA following CC BY NC 4.0 license.

**Instruction-Tuning Data**   See supplementary materials.

**Videos**   Available in https://websurg.com/, we provide the corresponding URL to each of the question-answer pair.

### A.2 PROMPTS

**Prompt for information extraction**   The prompt used to structurally extract key information from video title and transcript are in Figure 6.

```
Prompting Llama-3-70B to extract the structured information

messages = [ {"role":"system", "content": f"""You are an AI assistant specialized in surgical topics. You are
provided with a transcript from a surgical lecture video.
    Your task is to annotate the data in a structured way. Your target is to extract the meaningful visual
descriptions from the transcript.
    Your reply should follow the following json format:
    [{ "observation": "the observation includes the descriptions to surgical actions or organs, arteries, veins,
etc. from the transcript",
        "reason": "the reason or intention behind the observation if any, the reason of … is to …",
        "plan": "he surgical actions that can be performed if any, after …, we can …",
        "note": "notice about the observation, when …, note that …,"}]
    Below are requirements for the annotations:
    - Annotations may contain multiple observations and its corresponding reasons, plans, or notes.
    - Always use list to wrap the content even if there is only 1 observation.
    - Observation must only be descriptions to visible objects, actions.
    - Return an empty list if there is no descriptions to visible object or action in the transcript."""}
]
for sample in fewshow_samples:
    messages.append({"role":"user", "content":sample['context']})
    messages.append({"role":"assistant", "content":sample['response']})
messages.append({"role":"user", "content":'\n'.join(query)})
-----------------------------------------------------------------------------------------------------------
Part of Few-shot Examples
#1 input:
Yes. Yeah. So you have to do a good bite, trying to avoid the aorta.  There have been deaths reported after unsuspected puncture of the aorta with
secondary bleeding, rupture of the aorta.
#1 output:
[{
    "observation": "perform secure stitching",
    "note": "when perform the stiching you must avoid the aorta, since there have been deaths reported after
unsuspected puncture of the aorta with secondary bleeding, rupture of the aorta.",
    "plan": None,
    "reason": None
}]

#2 input:
A blue dye test was performed through a uterine manipulator cannula to check for tubal continuation and patency.
#2 output:
[{
    "observation": "a blue dye test was performed through a uterine manipulator cannula",
    "reason": "the reason of the blue dye test is to check for tubal continuation and patency",
    "note": None,
    "plan": None
}]
```

Figure 6: messages we use to prompt Llama-3-70B to extract structured information. query contains the transcribed text for each video clip and the video title.

**Prompt for question-answer generation for observation**    The prompt used to generate instruction data that describes a surgical video is in Figure 7.

```
Prompting Llama-3-70B to generate insturciton-tuning data for observation

messages = [ {"role":"system", "content": f"""You are an AI assistant specialized in surgical topics.
    You are provided with a text description of a surgical video clip from a surgical lecture. In some cases,
you may have additional text (title, description). Unfortunately, you don't have access to the actual video.
    Your task is to generate a Q&A pair or an answer to a given question about the video clip. The conversation
should proceed as though both the User and Assistant are viewing the video, while not referring to the text
information (title, description).
    Below are requirements for generating the questions and answers in the conversation:
    - Avoid quoting or referring to specific facts, terms, abbreviations, dates, numbers, or names, as these may
reveal the conversation is based on the text information, rather than the video clip itself. Focus on the visual
aspects of the video that can be inferred without the text information.
    - Do not use phrases like "mentioned", "title", "description" in the conversation. Instead, refer to the
information as being "in the video."""}]
for sample in fewshow_samples:
    messages.append({"role":"user", "content":sample['context']})
    messages.append({"role":"assistant", "content":sample['response']})
messages.append({"role":"user", "content":'\n'.join(query)})
```

Figure 7: messages we use to prompt Llama-3-70B to generate instruction-tuning data for observation. query contains the concatenated observations.

**Prompt for question-answer generation for reasoning**    The prompt used to generate instruction data for a variety of reasoning tasks is in Figure 8.

```
Prompting Llama-3-70B to generate insturciton-tuning data for reasoning

messages = [ {"role":"system", "content": f"""You are an AI assistant specialized in surgical topics.
    You are provided with a text description of a surgical video clip from a surgical lecture. In some cases,
you may have additional text (title, description). Unfortunately, you don't have access to the actual video.
    Your task is to generate a Q&A pair or an answer to a given question about the video clip. The conversation
should proceed as though both the User and Assistant are viewing the video, while not referring to the text
information (title, description).
    Below are requirements for generating the questions and answers in the conversation:
    - Avoid directly quoting or referring to specific facts, terms, abbreviations, dates, numbers, or names, as
these may reveal the conversation is based on the text information, rather than the video clip itself. Focus on
the visual aspects of the video that can be inferred without the text information.
    - Do not use phrases like "mentioned", "title", "description" in the conversation. Instead, refer to the
information as being "in the video."

    There can be four types of question, which are: reason which asks the reason of an action, plan which ask a
possible future step, note which asks for something you should notice when perform some action, and detail which
asks for more information about the observation,
    Generate a Q&A pair that you use the "statement" value to answer a question regarding the "observation".
    Your reply should be in the following json format: {"q": the_question, "a": the_answer, "type": qa_type}"""
for sample in fewshow_samples:
    messages.append({"role":"user", "content":sample['context']})
    messages.append({"role":"assistant", "content":sample['response']})
messages.append({"role":"user", "content":'\n'.join(query)})
-----------------------------------------------------------------------------------------------------------
Part of Few-shot Examples
#1 input:
Generate Q&A based on your understanding of the information below:
{
    "title": 'Laparoscopic Roux-en-Y gastric bypass for morbid obesity: a live educational procedure',
    "description": 'In this live educational video, Dr. Michel Vix demonstrates a stepwise laparoscopic Roux-en-Y gastric bypass procedure in a
39-year-old female patient with a BMI of 38. After stapled creation of the gastric pouch and splitting of the greater omentum, a stapled
(antecolic/antegastric) gastrojejunostomy and a jejunojejunostomy are performed. Both mesentery hernia ports are closed. ',
    "observation": 'there is a large left hepatic artery',
    "statement": 'if you have any traction here on your omentum, you have to stop and look if you have no adhesions that you need to open',
}
#1 output:
{
    "q": "What should you be aware of the omentum during this surgery?",
    "a": "You should be aware of if you have any traction here on the omentum, you have to stop and look if you
have no adhesions that you need to open",
    "type": "note"
}
```

Figure 8: messages we use to prompt Llama-3-70B to generate instruction-tuning data for reasoning. query provides a title, video description, observation, and statement to form a reasoning question-answer pair.

**Prompt for GPT evaluation** The prompt used to generate the evaluation results discussed in 5.2 is in Figure 9.

```
Prompting GPT-3.5 for question-answer evaluation

messages = [ {"role":"system", "content": f"""
You are a helpful and precise assistant for checking the quality of the answer.
        You will assign a score based on the correctness of the answer from 0 to 5. You will be given a list of
observations, and you will assign a score based on how well the answer matches the observations.
        - 5 means the answer correctly matches all obserations.
        - 4 means the answer correctly matches most obserations.
        - 3 means the answer correctly matches some obserations.
        - 1-2 means the answer only identifies the organs, instruments or actions.
        - 0 means the answer is completely wrong.
        - Assign a higer score for detailed explanations.
        - Do not assign higher score for too general answers, e.g. this is a laparoscopic surgery with out a
detailed type, manipluation of the organ without a detailed action or organ, use of instruments without a
detailed instrument.
        - reply in json format:
{"score": 0 to 5,  "acc": "(number of matched)/(number of total observations)", "comment": what are matched}"""
for sample in fewshow_samples:
    messages.append({"role":"user", "content":sample['context']})
    messages.append({"role":"assistant", "content":sample['response']})
messages.append({"role":"user", "content":'\n'.join(query)})
```

Figure 9: messages we use to prompt GPT-3.5 to evaluate models' output on surgical video questions.

Surgical Video Question-answer Example 209995552/212_216

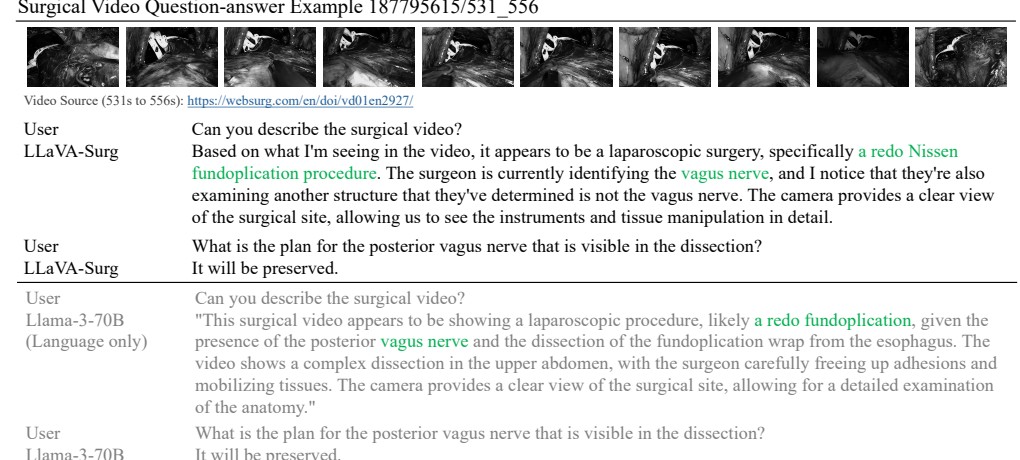

Video Source (212s to 216s): https://websurg.com/en/doi/vd01en3161/

| User | Can you describe the surgical video? |
|---|---|
| LLaVA-Surg | The surgical video appears to be showcasing an endoscopic procedure, specifically the extraction of a large stone from the bile duct. In the video, we see a balloon being positioned and then inflated to various pressures, which is likely being done to help retrieve the stone. The camera provides a close-up view of the procedure, allowing us to see the instruments and the stone itself. The video seems to be capturing a complex and delicate procedure, with the surgeon carefully manipulating the instruments to remove the stone. |
| User | Why is the controlled cholangiography performed in this procedure? |
| LLaVA-Surg | To confirm the absence of an intraductal stone. |
| User | Can you describe the surgical video? |
| Llama-3-70B (Language only) | The surgical video appears to be showcasing a minimally invasive procedure where the surgeon is performing a laparoscopic cholecystectomy, likely to remove the gallbladder. What's unique about this procedure is that it also involves an ERCP, or endoscopic retrograde cholangiopancreatography, which is a specialized procedure to examine the bile ducts. In the video, we see the surgeon using a balloon catheter to extract a stone from the bile duct, and also performing a controlled cholangiography, which involves injecting dye into the bile ducts to visualize them. The video provides a clear view of the surgical site, allowing us to see the instruments and the anatomy in detail. |
| User | Why is the controlled cholangiography performed in this procedure? |
| Llama-3-70B | To rule out the presence of stone in the common bile duct. |

Table 5: Example comparison of surgical video question-answering. We provided the response generated by the language-only Llama-3-70B, based solely on extracted information (transcript) for reference, it is considered the ground truth and represents the model's performance upper bound.

Surgical Video Question-answer Example 187795615/531_556

Video Source (531s to 556s): https://websurg.com/en/doi/vd01en2927/

| User | Can you describe the surgical video? |
|---|---|
| LLaVA-Surg | Based on what I'm seeing in the video, it appears to be a laparoscopic surgery, specifically a redo Nissen fundoplication procedure. The surgeon is currently identifying the vagus nerve, and I notice that they're also examining another structure that they've determined is not the vagus nerve. The camera provides a clear view of the surgical site, allowing us to see the instruments and tissue manipulation in detail. |
| User | What is the plan for the posterior vagus nerve that is visible in the dissection? |
| LLaVA-Surg | It will be preserved. |
| User | Can you describe the surgical video? |
| Llama-3-70B (Language only) | "This surgical video appears to be showing a laparoscopic procedure, likely a redo fundoplication, given the presence of the posterior vagus nerve and the dissection of the fundoplication wrap from the esophagus. The video shows a complex dissection in the upper abdomen, with the surgeon carefully freeing up adhesions and mobilizing tissues. The camera provides a clear view of the surgical site, allowing for a detailed examination of the anatomy." |
| User | What is the plan for the posterior vagus nerve that is visible in the dissection? |
| Llama-3-70B | It will be preserved. |

Table 6: Example comparison of surgical video question-answering. We provided the response generated by the language-only Llama-3-70B, based solely on extracted information (transcript) for reference, it is considered the ground truth and represents the model's performance upper bound.

## B    MORE EXPERIMENT RESULTS

We show more multimodal conversation examples in Table 5, 6, 7, 8.

---

Surgical Video Question-answer Example 213398446/50_79

Video Source (50s to 79s): https://websurg.com/en/doi/vd01en2509/

| | |
|---|---|
| User
LLaVA-Surg | Can you describe the surgical video?
This surgical video appears to be showcasing a laparoscopic procedure, likely a TME dissection, where the surgeon is working in the abdominal cavity. The camera provides a clear view of the surgical site, and we can see the surgeon using a cautery tool to carefully divide tissue, taking care to maintain a safe distance from a nearby major blood vessel. The video focuses on the dissection of a specific anatomical structure, with the surgeon working to separate and remove a block of tissue. |
| User
Llama-3-70B
(Language only) | Can you describe the surgical video?
This video shows a laparoscopic surgery where the surgeon is removing the rectum and a portion of the colon. The camera shows the surgeon dissecting and dividing tissue, and it appears that they are working in the lower abdominal area. The video focuses on the surgical site, with the surgeon's instruments and the tissue being operated on in clear view." |

Table 7: Example comparison of surgical video question-answering. We provided the response generated by the language-only Llama-3-70B, based solely on extracted information (transcript) for reference, it is considered the ground truth and represents the model's performance upper bound.

---

Surgical Video Question-answer Example 239796397/1065_1081

Video Source (1065 s to 1081s): https://websurg.com/en/doi/vd01en1556e/

| | |
|---|---|
| User
LLaVA-Surg | Can you describe the surgical video?
This surgical video appears to be showcasing a laparoscopic procedure, likely a trachelectomy and partial cystectomy, given the focus on the pelvic area and the dissection of the bladder from the uterine cervix. The video shows the surgeon carefully freeing the bladder from the uterus, and then retracting the uterine cervix to the left, providing a clear view of the operative site. |
| User
Llama-3-70B
(Language only) | Can you describe the surgical video?
This surgical video appears to be showing a laparoscopic procedure to treat a genitourinary prolapse. In the video, we see the urinary catheter in place, with the balloon positioned near the cervix. The surgeon is carefully maneuvering a prosthesis, taking care to avoid contact with the bladder and rectum. The camera provides a clear view of the pelvic area, allowing us to see the intricate anatomy and the surgeon's precise movements." |

Table 8: Example comparison of surgical video question-answering. We provided the response generated by the language-only Llama-3-70B, based solely on extracted information (transcript) for reference, it is considered the ground truth and represents the model's performance upper bound.

