# OpenReview forum: "LLaVA-Surg: Towards Multimodal Surgical Assistant via Structured Lecture Learning"
_ICLR.cc/2025/Conference — Submitted to ICLR 2025_

### Official Review · Reviewer_zRU3 · 2024-10-31

**Soundness:** 4
**Presentation:** 4
**Contribution:** 3
**Rating:** 6
**Confidence:** 4

**Summary:**

The paper introduces LLaVA-Surg, the first multimodal surgical assistant capable of understanding surgical videos and engaging in open-ended conversations about them. The authors create Surg-QA, a dataset of 102,000 surgical video-instruction pairs, using a novel two-stage question-answer generation pipeline. This approach reduces LLM hallucinations and costs by breaking down the generation process. The resulting model demonstrates superior performance in surgical video question-answering compared to previous general-domain models.

**Strengths:**

1. The pipeline is comprehensive: A two-stage question-answer generation process minimizes hallucinations by extracting information prior to generating pairs, which enhances data quality and reliability compared to Quilt-1M[1], which has a similar approach.

2. Integrating surgical visual concept alignment data through action triplets improves text-visual alignment, enhancing the model’s grasp of surgical concepts.

3. The idea is interesting: using the Spearman rank correlation between expert and GPT scores effectively validates the reliability of large-scale GPT evaluation.

[1] Ikezogwo, Wisdom, et al. "Quilt-1m: One million image-text pairs for histopathology." Advances in neural information processing systems 36 (2024).

**Weaknesses:**

1. Could you provide results for the three existing surgical-domain datasets (EndoVis-18-VQA, Cholec80-VQA, and SSG-VQA) trained on Surg-QA? These results could demonstrate Surg-QA's potential as a foundational dataset in the surgical domain.

2. Maybe considering to use other video VLM models, which provides a more sophisticated approach to temporal fusion than simple average pooling.

**Questions:**

Please address the weaknesses mentioned above.

---

### Official Review · Reviewer_QjG9 · 2024-11-01

**Soundness:** 2
**Presentation:** 2
**Contribution:** 3
**Rating:** 3
**Confidence:** 5

**Summary:**

The paper introduces a novel surgical multimodal dataset, which consists of over 102,000 video-instruction pairs generated through a two-stage pipeline, aimed at enhancing the understanding and conversational capabilities of surgical videos.

**Strengths:**

1. With over 102,000 video-instruction pairs, this dataset is the largest in the surgical field.
2. Structured data annotation pipeline using LLMs minimizes the risk of generating inaccurate or nonsensical content, improving dataset reliability.
3. Releasing the dataset, model, and code publicly fosters further research and development in the surgical AI domain.
4. The dataset can be a valuable resource for training and education, helping surgical trainees learn through interactive Q&A about real procedures.

**Weaknesses:**

1. The paper does not address how the data's quality is maintained as the videos are obtained from the web. The clinicians have reviewed the output of their MLLM model, but the paper does not confirm whether clinicians or domain experts have reviewed the raw data to ensure accuracy and reliability.
2. Concerns regarding the release, privacy, and permission risks associated with using sensitive surgical videos are not adequately discussed.
3. The paper lacks comprehensive validation across essential surgical downstream tasks and other surgical QA datasets, which are crucial for demonstrating clinical usability. There is also a need for more rigorous benchmarking against a broader range of state-of-the-art video MLLM architectures to establish the dataset's utility and the model's performance more robustly.
4. The comparison of the proposed methods with SOTA methods is limited and does not include the latest works. The manuscript also lacks evaluations with models trained on other surgical datasets, limiting the assessment of the proposed model's generalizability across different surgical scenarios.
5. The paper may need to evaluate the visual quality of the surgical videos.

**Questions:**

1. How can the quality of the data be ensured? The data collected may already contain a lot of noise and has been reprocessed by an LLM. Is there any person or clinician reviewing these raw data?
2. Can the data be released? Are there privacy and permission risks associated with the collected data?
3. The authors need to conduct more zero-shot evaluations on downstream tasks relevant to the surgical field, such as phase recognition, action/instrument classification, and other surgical domain VQA data to demonstrate the clinical usability of their method.
4. The authors need to compare with more state-of-the-art methods. The comparison methods in Table 3 were all first released in 2023.
5. The authors may verify their dataset on more benchmarks of SOTA Video MLLM architectures.
6. Also, the authors need more zero-shot comparisons with the same VLM trained on other surgical datasets, to showcase the generalizability of their proposed dataset.
7. The authors may evaluate the visual quality of the surgical videos themselves, as they are obtained from the website.

**Details Of Ethics Concerns:**

Potential copyright problem for online data.

---

### Official Review · Reviewer_6mWr · 2024-11-02

**Soundness:** 3
**Presentation:** 3
**Contribution:** 2
**Rating:** 5
**Confidence:** 4

**Summary:**

The paper introduces a novel vision-language model, LLaVA-Surg, designed to assist in surgical settings. Leveraging the newly created Surg-QA dataset with 102K surgical video-instruction pairs, the model provides conversational, open-ended responses to questions about surgical procedures. Evaluations demonstrate LLaVA-Surg’s superior performance in surgical video question-answering, indicating its potential as a reliable tool in surgery-related applications.

**Strengths:**

- The Surg-QA dataset, along with the two-stage pipeline, is a significant contribution to medical AI.
- LLaVA-Surg’s ability to process and interpret surgical video content sets it apart from other models focused primarily on static images.
- The language is clearly presented. The authors use precise and concise language so that the reader can easily understand the dataset, methodology, and results of the study.

**Weaknesses:**

- Although the dataset is valuable, this storyline and methodology is too similar with LLaVA-Med [1]. Maybe the authors could think of improvements of this simple fine-tuning method (i.e., SFT) to make better use of this dataset.
- The paper lacks comparative results. The current comparative models are rarely trained on surgical scene data, which is unfair. It is necessary to compare with a specific model.
- Since doctors are hired to do the annotation, have the possible ethical risks been resolved? For example, IRB approval, etc.

[1] Li C, Wong C, Zhang S, et al. Llava-med: Training a large language-and-vision assistant for biomedicine in one day[J]. Advances in Neural Information Processing Systems, 2023.

**Questions:**

- Improvement of the methodology.
- Detailed Comparison.

**Details Of Ethics Concerns:**

Since doctors are hired to do the annotation, have the possible ethical risks been resolved? For example, IRB approval, etc.

---

### Official Review · Reviewer_xkns · 2024-11-04

**Soundness:** 4
**Presentation:** 3
**Contribution:** 3
**Rating:** 5
**Confidence:** 4

**Summary:**

The paper introduces LLaVA-Surg, a multimodal conversational assistant based on surgical videos. Additionally, they introduce a new dataset with 102,000 question-answer pairs for training multimodal LLMs. The authors provide details of their data generation procedure, which is carefully designed to avoid hallucinations. The paper provides detailed comparisons with existing general-purpose and surgical-purpose datasets. Lastly, the authors provide a human and LLM evaluation of the dataset, showing consistent scores.

**Strengths:**

- **Clarity**: The paper is well-written and easy to follow.
- **Contributions**: This work makes a significant contribution to the development of surgical chat assistants. The dataset contains a wider range of surgical QAs compared to previous works. The proposed model and dataset may be valuable resources for researchers in this area.

**Weaknesses:**

- **Dataset Availability**: The surgical videos are available on WebSurg and are not a contribution of the authors. Therefore, the data availability may be subject to license changes from the content owners and WebSurg.
- **Hallucinations and Data Quality**: As the authors mentioned, there may be hallucinations in the dataset, since it is automatically generated. The authors provide chatGPT and human evaluations, but that is not enough to infer the data quality.
- **Model Availability**: It is not possible to reproduce the results since the model is not available yet, but enough details are provided to support the paper.

**Questions:**

The paper is very well written and addresses its objectives. It also supports its claims and provides adequate experiments. Therefore, I am leaning toward accepting this paper, but I have some minor concerns regarding the legality of using WebSurg's surgical videos. I also have some questions:
1. The authors mention that the model is limited by hallucinations, which is a serious concern for a surgical chatbot. Could you please provide more details, and types of hallucinations, and give some examples?
2. Would it be possible to evaluate LLaVA-Surg on the SSG-VQA dataset? I am interested in knowing more about the breadth of your dataset and if it contains enough information for cross-dataset generalization.

**Details Of Ethics Concerns:**

I am wondering about the WebSurg's policies on using their videos to train deep learning models, but I could not find any information about this in their terms of use.

---

### Official Review · Reviewer_A8vi · 2024-11-04

**Soundness:** 2
**Presentation:** 3
**Contribution:** 2
**Rating:** 5
**Confidence:** 4

**Summary:**

The paper introduces LLaVA-Surg, a multimodal large language model designed as a conversational assistant for surgical applications. To support this, the authors developed Surg-QA, a large-scale dataset containing 102,000 surgical video-instruction pairs, generated through a structured two-stage question-answer pipeline. This pipeline helps extract structured knowledge from surgical lecture videos, enabling the LLaVA-Surg model to understand complex surgical procedures and answer open-ended questions in a zero-shot setting. The model leverages CLIP for visual encoding and is fine-tuned on Surg-QA to specialize in surgical video question-answering, achieving superior performance compared to existing general-domain models.

**Strengths:**

1.	The authors provide a novel dataset, Surg-QA, which is a significant resource for training multimodal surgical models, covering diverse surgical procedures and question-answer pairs.
2.	The two-stage pipeline for question-answer generation mitigates hallucinations in LLM outputs, resulting in higher quality and reliability of generated data.
3.	LLaVA-Surg demonstrates notable improvements over general multimodal models in zero-shot surgical video question-answering tasks, showcasing its efficacy in understanding surgical context.

**Weaknesses:**

1. The paper should compare its model with recent multimodal LLM approaches, specifically ReAct (Yao et al., 2023), which combines reasoning and action for complex tasks.
[1] Yao, S., Zhao, J., Yu, D., Du, N., Shafran, I., Narasimhan, K., & Cao, Y. (2023, January). ReAct: Synergizing Reasoning and Acting in Language Models. In International Conference on Learning Representations (ICLR).
2. Using CLIP for frame-by-frame encoding lacks temporal modeling and increases processing costs and redundancy, burdening the LLM as frame count grows.
3. The paper lacks an in-depth error analysis, especially regarding potential hallucinations or misunderstandings in complex surgical scenarios. Although the authors claim to reduce hallucinations, achieving perfect performance seems challenging.
4. The model’s adaptability to other medical or clinical fields is unclear, as broader evaluations on datasets like RAD, SLAKE, and PathVQA are missing, which may limit its wider applicability.

**Questions:**

1. Does splitting video into frames for CLIP’s visual encoder lead to a loss of spatiotemporal information, and wouldn’t a video encoder like Video Swin Transformer [2] better capture temporal dynamics?
[2] Liu, Z., Ning, J., Cao, Y., Wei, Y., Zhang, Z., Lin, S., & Hu, H. (2022). Video swin transformer. In Proceedings of the IEEE/CVF conference on computer vision and pattern recognition (pp. 3202-3211).
2. How does LLaVA-Surg perform compared to other state-of-the-art multimodal methods? In addition to general multimodal models, a detailed comparison with models like ReAct would provide a more comprehensive evaluation. Has comparison with other two-stage methods [3] in VQA task been overlooked?
[3] Gai, X., Zhou, C., Liu, J., Feng, Y., Wu, J., & Liu, Z. (2024). MedThink: Explaining Medical Visual Question Answering via Multimodal Decision-Making Rationale. arXiv preprint arXiv:2404.12372.
3. Is the two-stage question-answer generation process applicable to other medical fields, and if so, what adjustments would be required? Additionally, validating the method’s performance on public datasets like RAD, SLAKE, and PathVQA would strengthen its generalizability.

---

### Meta-Review · Area_Chair_VxPb · 2024-12-15

**Metareview:**

This paper introduces Surg-QA, a large surgical video-instruction dataset, and LLaVA-Surg, a vision-language assistant for surgical video Q&A. The paper is well-written and presented clearly. It contributes a novel dataset, and strong results are achieved. However, key shortcomings of the paper include a lack of in-depth comparisons with state-of-the-art, concerns regarding the data quality, room for better temporal modelling, and reproducibility issues.

**Additional Comments On Reviewer Discussion:**

No rebuttal was provided and the reviewers kept or decreases their original scores.

---

### Decision · Program_Chairs · 2025-01-22

Reject